# Multifunctional Cell Regulation Activities of the Mussel Lectin SeviL: Induction of Macrophage Polarization toward the M1 Functional Phenotype

**DOI:** 10.3390/md22060269

**Published:** 2024-06-11

**Authors:** Yuki Fujii, Kenichi Kamata, Marco Gerdol, Imtiaj Hasan, Sultana Rajia, Sarkar M. A. Kawsar, Somrita Padma, Bishnu Pada Chatterjee, Mayuka Ohkawa, Ryuya Ishiwata, Suzuna Yoshimoto, Masao Yamada, Namiho Matsuzaki, Keita Yamamoto, Yuka Niimi, Nobumitsu Miyanishi, Masamitsu Konno, Alberto Pallavicini, Tatsuya Kawasaki, Yukiko Ogawa, Yasuhiro Ozeki, Hideaki Fujita

**Affiliations:** 1Graduate School of Pharmaceutical Sciences, Nagasaki International University, 2825-7 Huis Ten Bosch, Sasebo 859-3298, Japan; kawasakit@niu.ac.jp (T.K.); yogawa@niu.ac.jp (Y.O.); fujita@niu.ac.jp (H.F.); 2Department of Chemistry, KU Leuven, Celestijnenlaan 200G, 3001 Heverlee, Belgium; w175502f@yokohama-cu.ac.jp; 3Graduate School of Biomedical Sciences, Yokohama City University, 1-7-29, Suehiro, Tsurumi-Ku, Yokohama 230-0045, Japan; 4Department of Life Sciences, University of Trieste, Via Licio Giorgieri 5, 34127 Trieste, Italy; mgerdol@units.it (M.G.); pallavic@units.it (A.P.); 5Department of Microbiology, Faculty of Biological Science, University of Rajshahi, Rajshahi 6205, Bangladesh; hasanimtiaj@yahoo.co.uk; 6Department of Biochemistry and Molecular Biology, Faculty of Science, University of Rajshahi, Rajshahi 6205, Bangladesh; 7Center for Interdisciplinary Research, Varendra University, Rajshahi, Rajshahi 6204, Bangladesh; rajia_bio@yahoo.com; 8Department of Chemistry, Faculty of Science, University of Chittagong, Chittagong 4331, Bangladesh; akawsarabe@yahoo.com; 9Department of Oncogene Regulation Chittaranjan National Cancer Institute, 37 S.P. Mukherjee Road, Kolkata 700026, India; spadma03212@gmail.com (S.P.); cbishnup@gmail.com (B.P.C.); 10Graduate School of NanoBio Sciences, Yokohama City University, 22-2, Seto, Kanazawa-Ku, Yokohama 236-0027, Japan; ookawa.mayuka@aist.go.jp (M.O.); s192016g@yokohama-cu.ac.jp (R.I.); s202126b@yokohama-cu.ac.jp (S.Y.); yamada.mas.ug@yokohama-cu.ac.jp (M.Y.); s212104e@yokohama-cu.ac.jp (N.M.); s212115b@yokohama-cu.ac.jp (K.Y.); 148th.ave.seven@gmail.com (Y.N.); 11emukk LLC, 2-21-19, Matsunoki, Kuwana 511-0902, Japan; 12Graduate School of Food and Nutritional Sciences, Toyo University, 48-1, Oka, Asaka 351-8510, Japan; miyanishi@toyo.jp; 13National Institute of Advanced Industrial Science and Technology, Koto-Ku, Tokyo 135-0064, Japan; m-konno@aist.go.jp

**Keywords:** SeviL, R-type lectin, β-trefoil folding, asialo-GM1, M1 macrophage, cell proliferation, morphological alteration, Mytilidae

## Abstract

SeviL, a galactoside-binding lectin previously isolated from the mussel *Mytilisepta virgata*, was demonstrated to trigger apoptosis in HeLa ovarian cancer cells. Here, we show that this lectin can promote the polarization of macrophage cell lines toward an M1 functional phenotype at low concentrations. The administration of SeviL to monocyte and basophil cell lines reduced their growth in a dose-dependent manner. However, low lectin concentrations induced proliferation in the RAW264.7 macrophage cell line, which was supported by the significant up-regulation of TOM22, a component of the mitochondrial outer membrane. Furthermore, the morphology of lectin-treated macrophage cells markedly changed, shifting from a spherical to an elongated shape. The ability of SeviL to induce the polarization of RAW264.7 cells to M1 macrophages at low concentrations is supported by the secretion of proinflammatory cytokines and chemokines, as well as by the enhancement in the expression of IL-6- and TNF-α-encoding mRNAs, both of which encode inflammatory molecular markers. Moreover, we also observed a number of accessory molecular alterations, such as the activation of MAP kinases and the JAK/STAT pathway and the phosphorylation of platelet-derived growth factor receptor-α, which altogether support the functional reprogramming of RAW264.7 following SeviL treatment. These results indicate that this mussel β-trefoil lectin has a concentration-dependent multifunctional role in regulating cell proliferation, phenotype, and death in macrophages, suggesting its possible involvement in regulating hemocyte activity in vivo.

## 1. Introduction

Few lectins (carbohydrate-binding proteins) have been previously reported to induce lymphocyte differentiation by acting as growth factors [1,2,3]. For example, the GS-1 lectin from the legume *Griffonia simplicifolia* is an α-galactose-binding lectin that triggers the proliferation of T lymphocytes and induces macrophage polarization, exerting a cytotoxic activity against cancer cells [4]. On the other hand, the β-galactoside-binding lectin galectin-1 can bind glycans containing *N*-acetyllactosamine associated with T lymphocyte leukemia cells, inducing their apoptosis [5]. Other lectins were found to have cell-regulatory activities, being cytotoxic against cancer cells [6,7]. Among these, galactose/galactoside-binding lectins can bind to the glycans exposed on cell membranes, exerting regulatory effects by triggering an intracellular signaling cascade that induces differentiation, proliferation, and apoptosis [8,9,10]. In summary, the reciprocal interaction between lectins and cell-bound glycans is emerging as a prevalent yet mechanistically poorly understood process governing cellular proliferation, differentiation, and death.

SeviL is a β-galactoside-binding lectin that binds to GA1 (asialo-GM1: Galβ1-3GalNAcβ1-4Galβ1-4Glc) glycans, causing the apoptosis of cancer cells [11]. This lectin, first isolated from the gill of the mussel *Mytilisepta virgata*, has a polypeptide chain consisting of 129 amino acids and displays three tandem repeats of 40 amino acids each. The primary sequence of this lectin includes Q-X-W-like motifs, which are typically found in members of the R-type lectin family that adopt a β-trefoil folding [11,12]. SeviL is a dimeric lectin whose two subunits are non-covalently linked [12]. We have previously demonstrated that this lectin can regulate cell proliferation by activating the MAP kinase cascade following the recognition of glycosphingolipid ligands such as GA1 and GM1b (Neu5Acα2-3Galβ1-3GalNAcβ1-4Galβ1-4Glc) [11]. A study carried out on HeLa cells has shown that these events lead to the activation of caspases 3/9, which are the main molecular players in the induction of apoptosis. In light of the physiological roles played by lectins in Mollusca, this observed cytotoxicity may suggest a potential function of SeviL as a defense molecule.

This study revealed that administering low concentrations of SeviL to undifferentiated monocytic cells triggers their proliferation following the binding of the lectin to cell-membrane-associated glycans. Moreover, the phenotype of these cells was observed to undergo polarization toward inflammatory macrophages, which secrete various cytokines. This observation suggests that this mussel β-trefoil lectin exhibits multifunctional regulatory activities, which may vary depending on the target cell type and lectin concentration.

## 2. Results

### 2.1. SeviL Promoted the Proliferation of Macrophage Cell Lines

SeviL was previously demonstrated to regulate the fate of the cells that express GA1 glycans on their surface [11]. RAW264.7 cells are derived from macrophages expressing GA1 (Figure 1). In light of this observation, we evaluated whether SeviL treatment could affect immune cell growth (Figure 2); several different types of cells, such as the mouse macrophage cell line RAW264.7 (Figure 2, black bar), the human monocyte cell line THP-1 (white), and the rat basophil cell line RBL-1 (dark gray), were used for comparison. HeLa ovarian cancer cells (light gray) were also used as a positive control. Each cell line was cultured with SeviL (0–100 μg/mL) for 24 h, and their viability was measured using a WST-8 assay. The viability of all cell lines decreased significantly at concentrations exceeding 25 μg/mL. However, the viability of the RAW264.7 macrophage cell line increased after the administration of low concentrations (3–25 μg/mL) of SeviL. The cellular effects of SeviL and SeviL mutants were also compared (Figure 3). In contrast to dimeric SeviL (wild type), the monomeric SeviL [12] did not induce cell proliferation at 3–25 μg/mL in RAW264.7 cells. Fluorescent-labeled SeviL was detected around both the RAW264.7 and HeLa cells (a positive control) (Figure 4). However, only a small amount of SeviL was observed around the surface of THP-1 cells, which were not affected by the mitogenic activity of SeviL (Figure 2).

### 2.2. The Treatment of Macrophage Cells with SeviL Induces Morphological and Gene Expression Alterations

Transmission electron microscopy was used to investigate the morphology of the macrophage RAW264.7 cells before and after the administration of SeviL (10 μg/mL). Although RAW264.7 cells appeared spherical before the application of lectin (Figure 5A), incubation with SeviL for 24 h modified their cell shape, making it more elongated (Figure 5B). Since the observation of the cytosol indicated that the number of mitochondria increased (Figure 5C), the transcript level of mRNA encoding a major mitochondrial transporter (TOM22) was analyzed via RT-PCR. The expression of TOM22 was significantly up-regulated in a time-dependent manner with respect with the duration of incubation with the lectin (Figure 5D).

### 2.3. The Dual Activation of the JAK/STAT and MAPK Pathways in Macrophage Cells by SeviL

In the RAW264.7 cells treated with 10 μg/mL SeviL, the activation of both the JAK-STAT and MAPK kinase pathways occurred in a time-dependent manner (Figure 6). STAT-1 and STAT-3 were phosphorylated in RAW264.7 cells after administration of SeviL for 90 min. Interestingly, the phosphorylation of STAT-1 in RAW264.7 cells decreased 24 hours later. SeviL incubation also led to the phosphorylation of the extracellular-signal-regulated kinase, c-Jun N-terminal kinase, and P38 mitogen-activated protein kinase (Figure 3: p-Erk1/2 vs. Erk1; p-JNK vs. JNK; p-P38 vs. P38), which are classified as MAP kinases. After lectin administration, the activation of these kinases occurred more rapidly compared with the kinases involved in the JAK-STAT pathway. Indeed, the phosphorylation of Erk1/2, JNK, and P38 was already detectable after 15 min of incubation. Since no activation of the JAK-STAT pathway by SeviL was observed in THP-1 cells, the cell proliferation and polarization caused by SeviL in macrophage cell lines may be dependent on the dual involvement of the JAK-STAT and MAPK pathways. 

#### 2.3.1. SeviL Up-Regulates the Production of Cytokines and Chemokines in Macrophage Cells

SeviL administration induced the up-regulation of cytokines and chemokines in macrophage RAW264.7 cells. A protein array showed that SeviL induced the production of cytokines (IL-1α, IL-1rα, and IL-17) and chemokines (CCL-2, -3, -5, -22, and CXCL2) (Figure 7A). Furthermore, SeviL induced the expression of CD40, which is known to be expressed in antigen-presenting cells involved in acquired immunity. In addition, SeviL incubation also moderately induced the expression of TNF-α, CCL12, CXCL10, CXCL11, G-CSF, and lipocalin-2. Overall, the administration of SeviL to macrophage cells positively influenced the expression of several inflammatory molecules. These trends were markedly different from those observed when mussel lectins were added to cancer cells, since anticancer molecules, such as TNF-β, are produced to trigger apoptosis [14].

#### 2.3.2. Phosphorylation of Platelet-Derived Growth Factor Receptor (PDGFR)-α by SeviL Acts as a Trigger for Growing Macrophages

After RAW264.7 cells were incubated for 5 min with SeviL (10 μg/mL), the cells were harvested, and the lysate was extracted. Then, the lysates were subjected to a protein array analysis to detect the phosphorylation of tyrosine kinases. A strong phosphorylation of both PDGFR-α and MusK was observed after the incubation of SeviL (Figure 7B). Enhanced phosphorylation signals were also detected for a homolog of PDGFR, the FMS-like tyrosine kinase receptor (Flt-3), M-CSF-R, EGFR, and its homolog ErbB2 within 5 min after the treatment with SeviL. Moreover, tyrosine kinase C (TrkC) also displayed an increased phosphorylation level, and macrophage-stimulated protein receptors (MSPRs) were activated after the treatment with SeviL (Figure 7B).

### 2.4. The Overexpression of Multiple mRNA Markers Underlies M1 Macrophage Polarization following SeviL Treatment

The transcription of mRNAs encoding several markers that characterize the polarization of M1-stage macrophages was detected via RT-PCR after the administration of SeviL to RAW264.7 cells. The significant up-regulation of the transcripts encoding the cytokines and chemokines IL-1α, IL-6, TNF-α, CCL5, and the iron-sequestering protein lipocalin-2, well-known markers of M1 macrophage cells, was detected in RAW264.7 cell lines by treating SeviL for 3–12 h (Figure 8). In addition, the expression of the mRNA encoding inducible nitric oxide synthase (iNOS), which encodes another inflammatory marker of M1-stage macrophages, also increased at the same concentrations. On the other hand, neither IL-10, an anti-inflammatory cytokine, nor arginase-1, a marker of the M2 stage of macrophage cells, was differentially expressed in RAW264.7 cells after the treatment with SeviL. These results indicate that SeviL induced the polarization of RAW264.7 macrophages toward the M1 functional phenotype.

### 2.5. Cytokines and Chemokines Released into the Medium Are Regulated by the Lectin Concentration

An ELISA system was used to evaluate whether SeviL treatment could induce the polarization of RAW264.7 macrophage cells, altering the secretion of cytokines in the culture medium (Figure 9). The observed amount of secreted inflammatory-related cytokines IL-6 and TNF-α significantly increased even at low concentrations (2.5–10 μg/mL) of SeviL. On the other hand, the administration of monomeric SeviL or the co-presence of a haptenic sugar inhibited their secretion. These results support the ability of dimeric SeviL to stimulate the production of cytokines by polarized macrophages, even when the lectin is administered at low concentrations.

## 3. Discussion

Research carried out on plant-derived lectins such as ConA, PHA, PWM, VAA, jacalin, and banana lectin first reported the mitogenic activity of lectins, showing that these molecules could induce leukocyte differentiation and proliferation [15,16,17,18,19,20,21,22]. Similarly, the RAW264.7 macrophage cell line used in this study has been previously demonstrated to be susceptible to lectin-mediated regulation, since treatment with the PHA lectin could induce the polarization of these cells toward the M1 functional phenotype [23]. Furthermore, the ricin toxin B subunit has been shown to promote the phosphorylation of JAK/STAT molecules in this cell line [24].

Multiple examples of lectins displaying marked mitogenic properties isolated from different biological sources have been reported in the recent literature. One of the most relevant examples is represented by cyanovirin-N, a mannose-binding lectin of prokaryotic origin that can promote T cell proliferation [25]. In invertebrates, a GlcNAc-binding lectin from sponges stimulates the growth of monocyte cells [26], a mucin-binding lectin from gastropods can promote the proliferation of T cells [27], and an SUEL/RBL-type lectin from sea urchins has a similar effect on spleen cells [28]. C-type lectins isolated from marine invertebrates have also been found to have a mitogenic effect in T and B cells [29], as well as in macrophages [30]. Although many studies have reported the mitogenic activity of lectins in mammalian cells, several lectins are known to induce cell death if the treatment occurs in high concentrations [28,31]. By coincidence, we discovered the mitogenic activity of SeviL when its dose-dependent cytotoxic effect against blood-cell-derived culture cells was evaluated. Indeed, when SeviL was administrated to epithelial cells, basophil cells, monocytes and macrophage lineage cells, proliferation was inhibited at 50–100 μg/mL in all cases, with the lone exception of the macrophage RAW264.7 cell line. In this case, a significant increase in proliferation was observed at a concentration of 3–25 μg/mL (Figure 2). 

This result was similar to the proliferative effect previously reported for other lectins isolated from marine invertebrates in macrophage cells [30]. SeviL is a dimeric lectin with a β-trefoil structure that requires approximately 5 mM calcium for glycan binding. In contrast to dimeric SeviL, monomeric SeviL did not affect cell proliferation at any concentration, indicating that the mitogenic activity of this lectin, like its antitumor activity, requires dimerization (Figure 3). By exploiting the same RAW264.7 cell line previously used to study the proliferative effects of the sea cucumber C-type lectin CEL-I [31], we here provide a comparative analysis of the biological properties displayed by SeviL. Although these two lectins are not homologous and display a different tertiary structure, both of them bind to GalNAc, have a subunit molecular weight of 13–15 kDa, and require calcium to exert their carbohydrate binding activity with a dimeric structure [32]. However, they display different glycan binding properties: indeed, SeviL binds directly to the glycosphingolipid glycans GA1, GM1b, and SSEA-4, recognizing Galβ1-3GalNAcβ1-4Gal [11]. On the other hand, CEL-I binds to monosaccharides (GalNAc), GalNAcβ1-4Gal and GalNAcβ1-3Gal, which are the components of GM2 ganglioside and Gb4 globoside, respectively [30]. 

The expression of GA1 glycan, the ligand of SeviL, in RAW264.7 cells was confirmed in an immunohistochemical study (Figure 1). MytiLec-1, a Gal/GalNAc-binding lectin from a different Mytilidae species, which shares a β-trefoil structure with SeviL, was added with calcium to RAW264.7 cells. Nevertheless, this treatment had no significant effect on cell proliferation, indicating that the increase in the number of cells depends on specific lectin properties other than its 3D folding. SeviL induced the proliferation of RAW264.7 cells at a treatment concentration of approximately 3 μg/mL, but cell growth was inhibited at concentrations equal to 25 μg/mL or higher, mirroring the results obtained for adherent cells at the same concentrations. 

The property of inhibiting the proliferation of RAW264.7 cells at high concentrations marked another biological property shared by SeviL and CEL-I. SeviL stimulated RAW264.7 cells, triggering the overexpression of several pro-inflammatory cytokines (Figure 7) and promoting their polarization toward an M1 phenotype, mirroring the effects of CEL-I. NFκB and AP-1 are two rapidly inducible transcription factors, which act as master regulators of pro-inflammatory cytokine production [33]. Despite being subject to different regulatory pathways, they can modulate the expression of partly overlapping molecular targets, displaying a complex interplay with each other [34]. The observation that the production of several pro-inflammatory cytokines that are known to be directly regulated by both AP-1 and NFκB was significantly increased following SeviL treatment may therefore implicate the involvement of either of these two transcription factors (or both of them synergistically). However, while the marked activation of several kinases that are part of the MAPK cascade would definitely support a key role for AP-1, we could not collect any evidence to support a SeviL-mediated activation of NFκB. Indeed, we could not detect any down-regulation of IkappaBα (shown in Appendix A), a key regulator of NFκB activity that sequesters this transcription factor in its inactive form in the cytosol, releasing it only after its degradation, which is mediated by IκB kinases [35]. Regardless of the nature of their regulator, the secretion of IL-6 and TNF-α was suppressed by 40% when the presence of SeviL was combined with the addition of its inhibitory sugar lactose (Figure 9). This incomplete inhibition may be due to the insufficient inhibitory effect provided by this disaccharide [12], unlike the main ligand of SeviL, the glycosphingolipid glycan Galβ1-3GalNAcβ1-4Gal, which is a trisaccharide. Alternatively, the signal may be transmitted through protein–protein interactions between the 3D structure of SeviL dimers and the cell surface of RAW264.7 cells [36]. 

The importance of similar protein–protein interaction processes in mediating cytokine secretion was also reported in previous studies, such as those concerning the polarization of RAW264.7 cells following CEL-I treatment [30,37]. On the other hand, the observation that the granulocyte colony-stimulating factor (G-CSF) is secreted in response to SeviL, as previously reported for CEL-I [30], is particularly noteworthy. This finding reveals, from a comparative biochemical perspective, that different lectins from marine invertebrates can regulate the production of key molecules involved in macrophage polarization. In addition to this stimulatory activity, the administration of a low concentration of this bivalve lectin also led to significant morphological changes in RAW264.7 cells, which switched from a round to an elongated shape (Figure 5). In addition, a quantitative PCR analysis revealed an eight-fold increase in the transcription of the mitochondrial gene Tom22 compared with that in control RAW264.7 cells the 12 h of treatment with SeviL. Alterations in the intracellular signaling pathway of SeviL-treated RAW264.7 cells were also revealed by subsequent experiments. Indeed, after just 15 min after the administration of SeviL, the level of phosphorylation of different intracellular MAP kinases, including the mitogen-activated kinase JNK, p38 mitogen-activated kinase, and extracellular-signal-regulated kinase 1/2, significantly increased. A longer treatment (i.e., 90 min) was necessary to determine the activation of the cytokine receptor JAK/STAT in RAW264.7 cells (Figure 6), which may therefore indicate that the activation of this pathway was not directly due to SeviL, but rather secondarily triggered by the up-regulation of several cytokines regulated by the treatment with the lectin. 

We hypothesize that the increased production of G-CSF and IL-6 is due to the polarization of RAW264.7 cells, which seems to be supported by the increased phosphorylation of the tyrosine kinases activated following the binding of G-CSF to its receptor, as observed through the antibody array (Figure 7). SeviL activated the intracellular MAP kinase system and stimulated the secretion of cytokines such as IL-1α and TNF-α. Subsequently, these cytokines most likely led to the production of nitric oxide synthase (iNOS) and cyclooxygenase (COX2). After treating RAW264.7 cells with SeviL, the timing at which the expression of different genes increased was remarkably different. One of the earliest responsive genes was IL-1α, followed by TNF-α, iNOS, COX2, the iron-sequestering protein lipocalin-2, and the chemokine CCL5. These results suggest that all these molecules are involved in a standard regulatory network aimed at ultimately inducing the polarization of macrophages toward a pro-inflammatory functional phenotype. Cell surface binding of SeviL on RAW264.7 cells was also observed via a confocal microscopic analysis (Figure 4). This observation suggested that this binding triggers the activation of an intracellular signaling cascade, which is ultimately responsible for the observed morphological cellular alterations. Taken together, the results of these experiments show that SeviL possesses mitogenic activity toward RAW264.7 cells, which can lead to their polarization into M1 macrophages through the production of inflammatory-related molecules. 

However, it is essential not only to emphasize the cell surface glycan-dependent cytotoxic mechanism, as reported for the anticancer effects of the bivalve β-trefoil lectins MytiLec-1 and SeviL, but also to investigate whether lectin interactions with cell surface proteins are relevant for comprehending the cell regulatory system they control. Mitogenic activities similar to those observed for SeviL have been previously reported for other lectins, such as those from the bivalve *Tridacna maxima* and the gastropod *Belamyia bengalensis* [27,38,39]. Recently, peptides obtained by degrading proteins from pearl-producing and edible bivalves have been found to regulate the function of RAW264.7 cells [40]. These compounds suppressed the expression of proinflammatory cytokines (TNF-α, IL-1), iNOS, and COX2, the expression of which was found to be increased in this study by the addition of SeviL. However, these molecules also led to the phosphorylation of ERK1/2, JNK, and p38, mirroring the results obtained with SeviL. 

The SeviL molecule used in this study was genetically engineered and has no peptide degradation product contamination. Marine lophotrochozoans, which lack acquired immunity, have developed an efficient immune system to defend themselves against the high microbial loads associated with seawater. Hemocytes, the main cellular players in molluscan immunity, share many features with vertebrate macrophages and can be considered as their prototypes [41,42]. Within this context, the physiological function of a lectin with properties such as SeviL might be to regulate the activity of hemocytes. The presence of homologs of MAP kinases has been confirmed in mussels via transcriptome analysis. Although the nature of most cytokines in bivalves remains elusive, these organisms most certainly produce signaling molecules that are functionally analogous to their vertebrate counterparts [43,44]. Unfortunately, methods for establishing primary cultures of mollusk cells have not yet been established. Nevertheless, intense research has been carried out on methods for the primary culture of blood cells [45,46]. Combining the knowledge about the mitogenic mechanisms exerted by lectin–glycan interactions on macrophages with a comparison of phagocytic cells between mollusks and mammals would undoubtedly provide a more comprehensive understanding of the role played by these molecules in the context of marine mollusk immune defenses. 

## 4. Materials and Methods

### 4.1. Materials

RAW264.7 (mouse macrophage), THP-1 (human monocyte), RBL-1 (rat basophil), and HeLa (human epithelial cancer) cell lines were obtained from the JCRB cell bank (Ibaraki, Osaka, Japan). Ethanol, chloroform, lactose, cell lysis buffer (20 mM Tris-HCl, pH7.4, 200 mM NaCl, 2.5 mM MgCl_2_, 0.05w/v% NP-40 substitute, 0.05% sodium azide), stripping solution, trypsin for cell culture, RPMI 1640 medium, fetal bovine serum (FBS), penicillin–streptomycin solution, horseradish peroxidase (HRP)-conjugated β-actin monoclonal antibody (mAb), and the LBIS Mouse TNF-α ELISA kit were all acquired from FUJIFILM Wako Pure Chemical Corp. (Osaka, Japan). The lactosyl–agarose gel was obtained from EY Laboratories (San Mateo, CA, USA). Standard protein markers for SDS-PAGE were purchased from Takara Bio Inc. (Kyoto, Japan). Paraformaldehyde and HRP-conjugated donkey anti-mouse IgG were obtained from Merck KGaA (Darmstadt, Germany). Blocking solution, phosphatase inhibitor cocktail tablets, and protease inhibitor cocktail were obtained from Nacalai Tesque Inc. (Kyoto, Japan). Anti-P38 mAbs, anti-phosphorylated P38 (pT180/pY182) mAbs, anti-Erk1 and anti-phosphorylated ERK_1/2_ (pT202/pY204) mAbs, and anti-SAPK/JNK and anti-phosphorylated JNK/SAPK1 (pT183/pY185) mAbs were procured from Becton Dickinson (Franklin Lakes, NJ, USA). The anti-phosphorylated STAT-1 mAb was obtained from Abcam (Cambridge, UK), whereas the anti-phosphorylated STAT-3 mAb was acquired from Cell Signaling Technology (Danvers, MA, USA). Direct-zol^TM^ RNA MiniPrep was provided by Zymo Research (Orange, CA, USA). Trizol reagent and the Pierce^TM^ BCA protein assay kit were obtained from Thermo Fischer Scientific (Waltham, MA, USA). THUNDERBIRD^TM^ Next SYBR qPCR Mix, ReverTra Ace, and Can Get Signal Immunoreaction Enhancer Solutions 1 and 2 were obtained from Toyobo Co. (Osaka, Japan). The 25% glutaraldehyde solution, EPON 812 resin, DMP-30, and DDSA-EM were purchased from TAAB laboratories (Aldermaston, UK). The Cell-Counting Kit-8 (including WST-8[2-(2-methoxy-4-nitrophenyl)-3-(4-nitrophenyl)-5-(2,4-disulfophenyl)-2H-tetrazolium monosodium salt]) and HiLyte Flour^TM^ 555 labeling kit—NH_2_ were provided by Dojindo Laboratories (Kumamoto, Japan). PVDF membranes for electroblotting and the peroxidase substrate EzWestBlue were acquired from ATTO Corp. (Tokyo, Japan). The proteome profiler mouse cytokine array kit and proteome profiler mouse phosphor-RTK array kit were obtained from (Minneapolis, MN, USA). The Ray Bio Mouse IL-6 ELISA Kit was purchased from Ray Biotech (Peachtree Corners, GA, USA).

### 4.2. Protein Expression of SeviL and Its Mutant

The lectin was purified according to a previously described procedure [12]. pET28-SeviL was transformed into *E. coli* BL21 (DE3) cells, which were subsequently grown at 310 K with shaking in 6 L of LB medium containing kanamycin (30 μg/mL) and chloramphenicol (20 μg/mL). When the culture showed O.D. 0.6–0.7 at 600 nm, the SeviL expression was induced by adding IPTG to a final concentration of 0.5 mM, and growth was continued for 3 h at 37 °C. The cells were collected via centrifugation at 3000× *g* at 4 °C for 30 min. The pellet was suspended in 100 mM Tris HCl, pH 8.0, and 150 mM NaCl, and then lysed by sonication on ice. The lysate was centrifuged at 38,000× *g* at 4 °C for 45 min. The supernatant solution was loaded onto a 10 mL lactosyl–sepharose column (EY Laboratories). After washing, it was eluted with 20 mM Tris HCl buffer (pH 8.0, 150 mM NaCl) containing 100 mM lactose sugar. The significant protein fraction was collected and dialyzed overnight to obtain purified SeviL. The monomeric SeviL(Q12R/F126K) was expressed according to the previous work [12].

### 4.3. Cell Viability and Cell Surface Staining

The Cell-Counting Kit-8 (CCK-8) containing WST-8 was used to evaluate cell viability [43]. The immune cell lines RAW264.7, THP-1, and RBL-1 were cultured and maintained in RPMI 1640 supplemented with heat-inactivated FBS (10%, *v*/*v*), penicillin (100 IU/mL), and streptomycin (100 μg/mL) at 37 °C in a 95% air/5% CO_2_ atmosphere. The experimental samples were divided into the untreated control and SeviL and monomeric SeviL-treated groups. The cells were treated with 3–100 μg/mL of SeviL or monomeric SeviL for 24 h, WST-8 was added, and the cells were cultured for 4 h. The optical density was measured using a microplate spectrophotometer at 450 nm.

The cell surface was stained using fluorescence-labeled (λex/em 555/570 nm) SeviL with a HiLyte Fluor 555 labeled kit—NH_2_ as described in a previous method [11]. The cells (1 × 10^6^) were fixed with 4% paraformaldehyde in PBS for 15 min, washed 3× with PBS, and blocked with 1% BSA in PBS for 30 min at room temperature. The cells were washed 3× with PBS, incubated with fluorescence-labeled SeviL (diluted 1:200 with PBS) at 4 °C for 2 h, and washed 3× with PBS. Cells were placed onto low-fluorescence glass slides, mounted with 50% glycerol solution, and examined via confocal microscopy. Confocal images were obtained using an FV10i FLUOVIEW (Olympus, Tokyo, Japan).

### 4.4. Detection of Activated Signal Transduction Molecules in RAW264.7 Cells

The murine macrophage cell line RAW264.7 (3 × 10^5^ cells) was cultured with SeviL (0–100 μg/mL) for 24 h and lysed with 200 μL of cell lysis buffer. The cell lysates were separated via SDS-PAGE and electroblotted onto the PVDF membranes. The primary antibodies used were directed against P38 (mouse mAb; dilution 1:3000), phospho-P38 (mouse mAb; dilution 1:3000), p-Erk_1/2_ (mouse mAb; dilution 1:3000), Erk_1/2_ (mouse mAb; dilution 1:3000), p-JNK (mouse mAb; dilution 1:3000), JNK (mouse mAb; dilution 1:3000), STAT-1 (mouse mAb; dilution 1:3000), and STAT-3 (rabbit mAb; dilution 1:3000). The membrane was masked with blocking solution at RT, incubated with HRP-conjugated goat anti-mouse IgG (for mouse mAb) or anti-rabbit IgG (for rabbit mAb) for 1 h [12], and detected with EzWestBlue. The experiments were performed in triplicate.

### 4.5. Protein Array

A proteome profiler mouse XL cytokine array kit (R&D systems, MN, USA) and a proteome profiler mouse Phospho-Receptor Tyrosine Kinase (RTK) array kit (R&D systems, MN, USA) were used to compare the cellular reactions between SeviL-treated and untreated macrophage cell lines. Each cell line was lysed, and the protein concentration was measured using the BCA method [47]. The detection of cytokines and phosphorylated RTK in RAW264.7 macrophage cell lines was performed according to the manufacturer’s instructions. The membranes were blocked by incubation with the blocking buffer at room temperature for 30 min. Three hundred milligrams of each sample was applied to the membrane and incubated at 4 °C overnight. The membranes were washed 3 times with washing buffer at room temperature for 5 min per wash and incubated with biotin-conjugated antibodies at room temperature for 90 min. Finally, the membranes were washed and incubated with horseradish peroxidase-conjugated streptavidin at room temperature for 1 h. A ChemiDoc imaging system was used to detect signals (Bio-Rad Laboratories, Inc., Hercules, CA, USA).

### 4.6. Detection of Cytokines by ELISA

The levels of IL-6 and TNF-α in cell culture medium (mouse IL-6 ELISA Kit, Proteintech, Japan and mouse TNF-α ELISA Kit, Fujifilm Wako chemicals, Tokyo, Japan) were measured via ELISA according to the manufacturer’s instructions. The medium was diluted from 1:50 to 1:100. Samples (100 μL) were added to the ELISA plate separately and incubated for 2 h at room temperature. After the usual procedure, biotinylated antibodies were added and the samples were incubated for 60 min at room temperature. After washing 4 times, the conjugate (concentrated HRP conjugate/HRP conjugate diluent = 1:100) was added and incubated for 30 min at room temperature. After washing the plate 4 times, the substrate reagent (TMB) was added, and the plate was incubated for 15 min at room temperature in the dark. A termination solution was added to stop the reaction. The optical density of each well was measured in a microplate reader (Molecular Devices Japan K.K, Tokyo, Japan) at a wavelength of 450 nm. All the samples were measured in triplicate.

### 4.7. RNA Isolation and Quantitative Real-Time PCR (RTPCR)

The expression of key genes related to inflammation, i.e., iNOS, IL-6, TNF-α, IL-1-α, COX-2, lipocalin-2, and CCL-5, was detected via quantitative and RT-PCR methods. According to the manufacturer’s protocol, total RNA was extracted from RAW264.7 cells (5 × 10^5^) using Trizol (Life Technologies Japan Ltd., Tokyo, Japan). RNA was quantified using a NanoDrop (Thermo Scientific) with ND-1000 software. cDNA synthesis was performed using ReverTra Ace (Toyobo Co. Ltd., Osaka, Japan). Primers for each gene were obtained with NCBI gene pick (http://www.ncbi.nlm.nih.gov, accessed on 20 October 2020). The expression levels of each gene were detected via quantitative RT-PCR on a 7900HT fast real-time PCR system (excitation at 483 nm and emission at 553 nm) (Thermo Fisher Scientific, Waltham, MA, USA) using the THUNDERBIRD Next SYBR qPCR Mix (Toyobo Co., Ltd., Osaka, Japan). Measurements were normalized to endogenous β-actin levels. Relative fold changes in expression were calculated using the ΔΔCT method. The primer sequences for iNOS, IL-6, TNF-α, IL-1α, COX-2, lipocalin-2, CCL5, IL-10, and arginase-1 were as follows: iNOS, 5′-GAGGCCGCATGAGCTTGGTGTTT-3′ (forward) and 5′-GGGGGTTGCATTTCGCTGTCTCC-3′(reverse); IL-6, 5′-TCCGGAGAGGAGACTTCACA-3′ (forward) and 5′-CATAACGCACTAGGTTTGCCG-3′ (reverse); TNF-a, 5′-CCCTCACACTCAGAT CATCTTCT-3′ (forward) and 5′-GCTACGACGTGGGCTACAG-3′ (reverse); IL-1a, 5′-AGGGAGTCAACTCATTGGCG-3′ (forward) and 5′-TGGCAGAACTGTAGTCTTCGT-3′ (reverse); COX2, 5′-CCTGCTGCCCGACACCTTCAACAT-3′ (forward) and 5′-CAGCAACCCGGCCAGCAATCT-3′ (reverse); lipocalin-2, 5′-TCTGTCCCCACCGACCAAT-3′ (forward) and 5′-GGAAAGATGGAGTGGCAGACA-3′ (reverse); CCL5, 5′-TTGCCTACCTCTCCCTCGC-3′ (forward) and 5′-TCGAGTGACAAACACGACTG-3′ (reverse); IL-10, 5′-CGGGAAGACAATAACTGCACCC-3′ (forward) and 5′-CGGTTAGCAGTATGTTGTCCAGC-3 (reverse); arginase-1, 5′-CATTGGCTTGCGAGACGTAGAC-3′ (forward) and 5′-GCTGAAGGTCTCTTCCATCACC-3′ (reverse); Tom-22, 5′-GACGACGACGACGAGCTAGA-3′ (forward) and 5′-CCTCTCGGGAAACATCTCCG-3′ (reverse); and β-actin, 5′-GGAATGGGTCAGAAGGACTC-3′ (forward) and 5′-CATGTCGTCCCAGTTGGTAA-3′ (reverse). 

### 4.8. Transmission Electron Microscopic Analysis of Macrophage Cells

SeviL-treated and untreated RAW264.7 cells were fixed with 4% paraformaldehyde and 0.2% glutaraldehyde and kept in the same fixative for one hour at 4 °C. The cell pellets were dehydrated by a graded ethanol series, soaked in propylene oxide at r.t., embedded in EPON resin, and then polymerized under heat at 60 °C. Thin sections were cut by a diamond knife on a Reichert Ultracut R microtome (AMETEK Reichert, Depew, NY, USA) and mounted on nickel grids. Sections were contrasted with uranyl acetate and lead citrate and examined via transmission electron microscopy (model H7650, Hitachi; Tokyo) at an acceleration voltage of 80 kV.

### 4.9. Statistical Analysis

The experiments were performed in triplicate, and the results are presented as means ± standard error (SE). The data were subjected to a one-way analysis of variance (ANOVA) followed by Dunnett’s test using the SPSS Statistics software package, v. 10 (www.ibm.com/products/spss-statistics, accessed on 5 February 2021). Differences with *p* < 0.05 were considered significant.

## 5. Conclusions

In a previous study, we demonstrated that SeviL triggers glycan-dependent cell death in asialo-GM1-positive cancer cells by activating MAP kinase and caspases. However, the addition of SeviL at low concentrations to a macrophage cell line had a markedly different effect, inducing the proliferation, activation, and polarization of M1 macrophage cells. As some toxins can be turned into valuable drugs in small doses, this suggests that the otherwise cytotoxic SeviL can activate macrophage cell effects at low concentrations. In mussels, such as *M. virgata,* hemocytes represent the quintessential players in the innate immune response, protecting these bivalves from the threat of invading microbes in challenging marine coastal environments. These multifunctional circulating cells are well known to produce cytotoxic effectors and inflammatory cytokines, which coordinate cellular and humoral responses, aiding the elimination of pathogens. Moreover, hemocytes display a marked phagocytic activity, which clearly indicates striking functional similarities with vertebrate macrophages. Considering these observations, one might wonder whether SeviL is involved in regulating the proliferation of hemocytes. The establishment of a standardized, reproducible method for culturing mussel hemocytes will provide further information to clarify the physiological function of this marine lectin.

## Figures and Tables

**Figure 1 marinedrugs-22-00269-f001:**
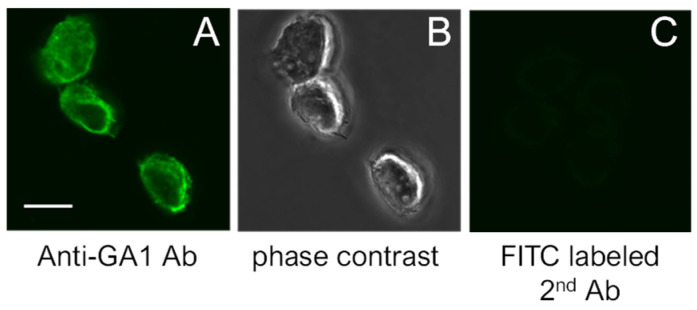
(**A**): Expression of GA1 on the surface of RAW264.7 cells. (**B**): Phase contrast views of (**A**). (**C**): Negative control. The scale bar indicates 10 μm.

**Figure 2 marinedrugs-22-00269-f002:**
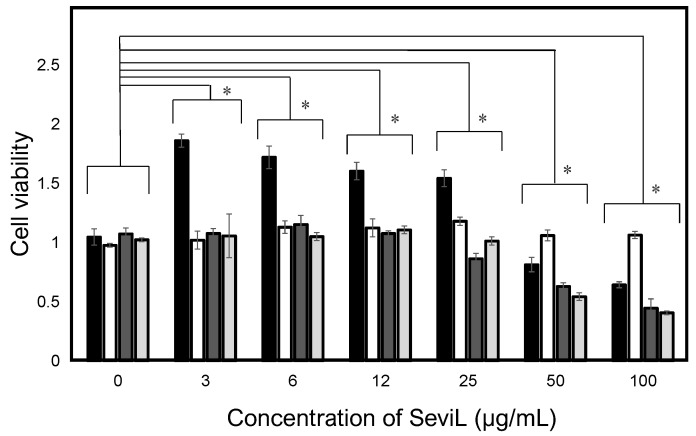
Cellular effects of SeviL. The black, white, dark gray, and light gray bars indicate the mouse macrophages (RAW264.7), human monocytes (THP-1), rat basophils (RBL-1), and human ovarian cancer (HeLa) cells, respectively. Cells were treated with SeviL at various concentrations (0–100 μg/mL^−1^) for 24 h, and cell viability (expressed as A_450_; see experimental design: cell viability and cytotoxicity assays) was determined via a WST-8 assay. The data shown are means ± SE (*n* = 3). *p*-values (* *p* < 0.05).

**Figure 3 marinedrugs-22-00269-f003:**
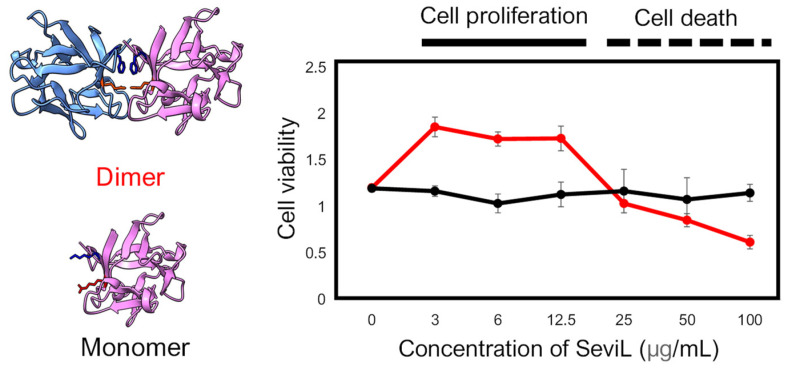
Comparison of the cellular effect observed in the RAW264.7 cell line between dimeric SeviL (wild type: left upper and right red line) versus monomeric SeviL (mutant: left lower and right black line). Each point represents an average of triplicate measurements. Molecular graphics showing the 3D structures of SeviL were generated by ChimeraX [13].

**Figure 4 marinedrugs-22-00269-f004:**
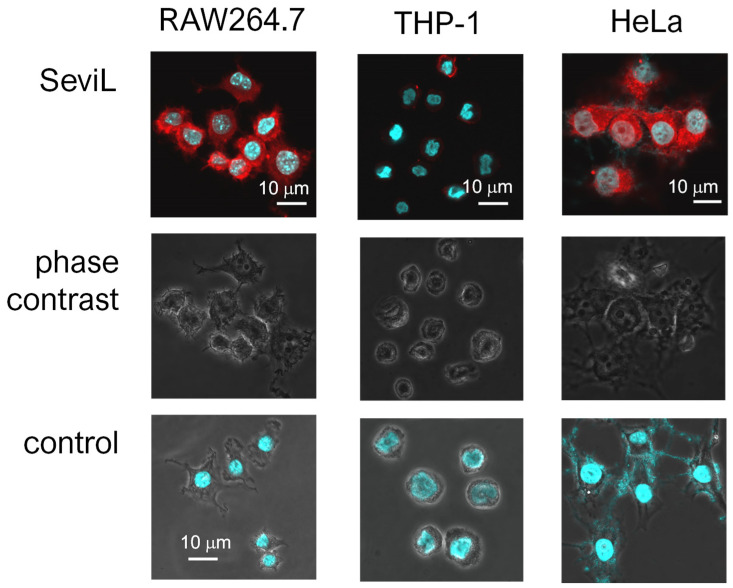
Different affinities of SeviL for cell membranes. Paraformaldehyde-fixed cells were observed via phase contrast and luminescent scanning microscopy. Staining was performed with fluorescent-labeled SeviL (red) and Hoechst (blue). Scale bar: 10 μm. Controls indicate only Hoechst staining of each cell line.

**Figure 5 marinedrugs-22-00269-f005:**
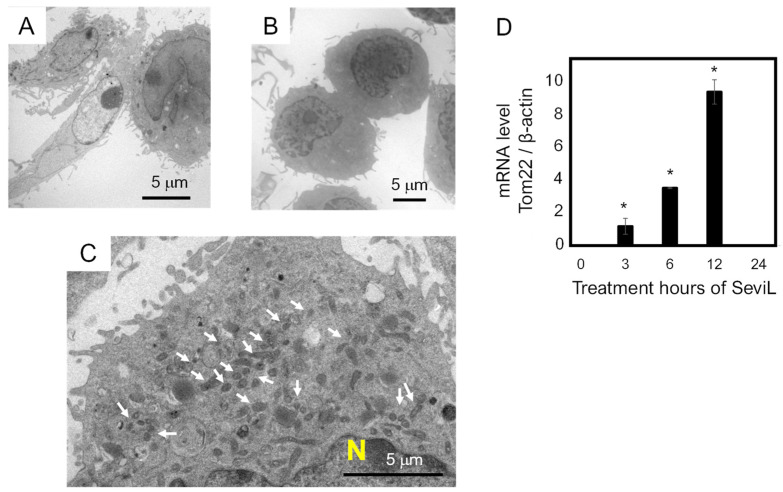
Morphological changes in RAW264.7 macrophage cells observed via transmission electron microscopic analysis. Panel (**A**) shows the morphological features of RAW264.7 cells. Panels (**B**,**C**) show the RAW264.7 cells after treatment with SeviL. The arrows in panel C show an increase in the number of mitochondria. Panel (**D**) shows the mRNA level of Tom22 quantified via quantitative PCR. The scale bars in (**A**–**C**) indicate 5 μm. The data shown are the means ± SE (*n* = 3). *p*-values (* *p* < 0.05).

**Figure 6 marinedrugs-22-00269-f006:**
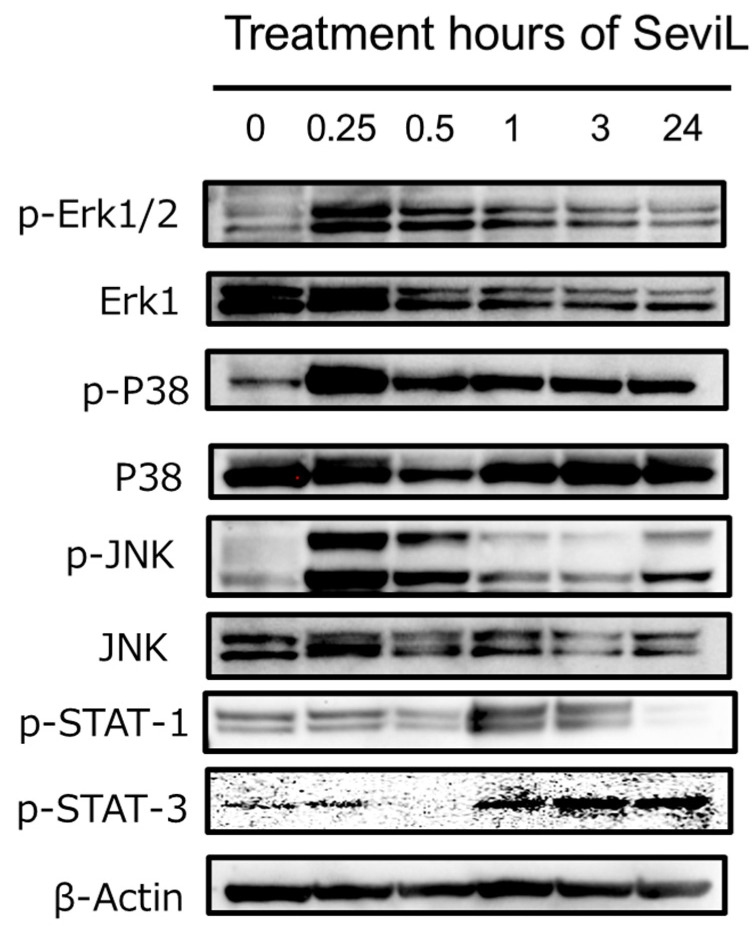
Phosphorylation of JAK/STAT and MAPKs by SeviL in RAW264.7 cells. Cells (5 × 10^5^) were treated with SeviL (10 μg/mL^−1^) over several time frames (0–24 h). The phosphorylation of signaling molecules in the cell lysates was evaluated via Western blotting. p-JNK, p-STAT-1, p-STAT-3 p-Erk_1/2_, and p-P38 are the phosphorylated forms of JNK, STAT-1, STAT-3, Erk_1/2_, and P38, respectively.

**Figure 7 marinedrugs-22-00269-f007:**
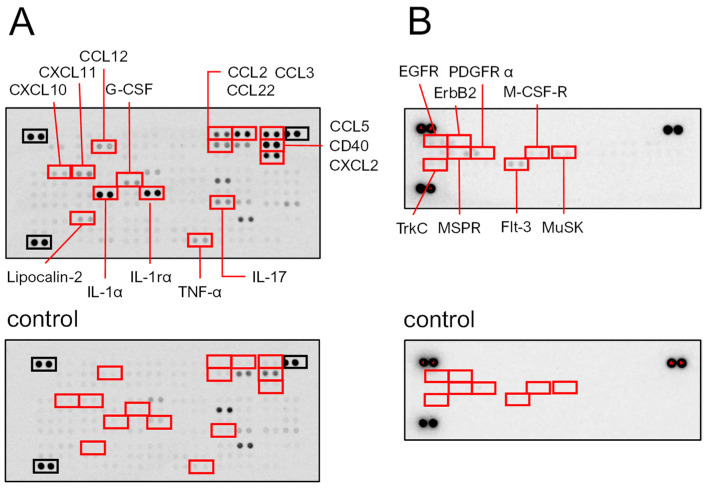
Macrophage RAW264.7 cells were cultured with or without SeviL (10 μg/mL) for 24 h, and each lysate was applied to a proteome profiler mouse cytokine array (**A** and **A** control). The cells were cultured with or without SeviL (10 μg/mL) for 5 min, and each lysate was applied to a receptor tyrosine kinase antibody array (**B** and **B** control). The red boxes indicate the proteins whose expression differed significantly between the SeviL-treated and untreated cells in the panel. The black boxes indicate positive controls.

**Figure 8 marinedrugs-22-00269-f008:**
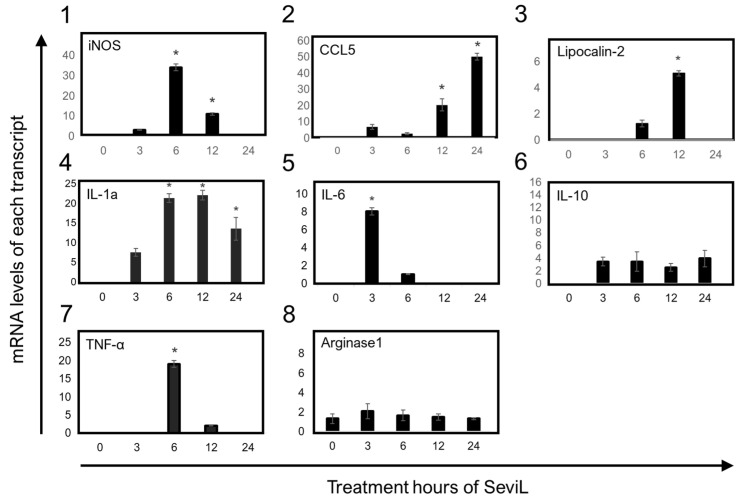
The effects of SeviL on inflammatory responses in RAW264.7 cells. The effect of the treatment with SeviL was evaluated at several time points (0, 3, 6, 12, and 24 h). The gene expression levels of iNOS, IL-6, TNF-α and IL-1α, IL-10, arginase-1, lipocalin-2, and CCL-5 in RAW264.7 cells were quantified via quantitative real-time PCR. Compared with the control group, * *p* < 0.05.

**Figure 9 marinedrugs-22-00269-f009:**
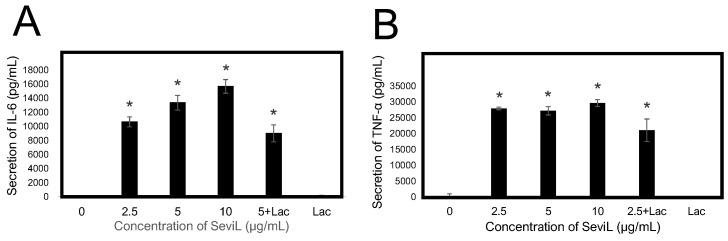
Effect of SeviL on RAW264.7 cells induced the secretion of cytokines. Cells were cultured without or with various concentrations of SeviL (0–10 µg/mL) for 24 h in the presence or absence of 10 mM of lactose, the haptenic sugar of SeviL. Cytokine, IL-6 (**A**), and TNF-α (**B**) levels in the supernatant were measured via ELISA. The data shown are means ± SE (*n* = 3). Compared with the control group, * *p* < 0.05.

## Data Availability

The data presented in this study are available on request from the corresponding author upon reasonable request.

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
