# Peer review of "Multifunctional Cell Regulation Activities of the Mussel Lectin SeviL: Induction of Macrophage Polarization toward the M1 Functional Phenotype"

_marinedrugs, 2024, doi:10.3390/md22060269_

Round 1

Reviewer 1 Report

Comments and Suggestions for Authors

The manuscript entitled “Multifunctional cell regulation activities of the mussel lectin SeviL: the differentiation into M1 macrophages” by Yuki Fujii et al described the Sevil stimulatory effects on a murine macrophage cell line Raw264.7. It provides some insight of Sevil, however, many concerns have been raised.

Major:

1.     In general, M-CSF and GM-CSF are major macrophage differentiation factors; M1, M2 terminology are linked with macrophage polarization status, not their differentiation.

2.     JAK/STAT pathway can be activated by many cytokines, including the most potent anti-inflammatory cytokine IL10. The activation of STATs, especially STAT3, cannot be directly linked to proinflammatory function of Sevil.

3.     As NFkB is a major player in the expression of proinflammatory cytokines/chemokines, its activity upon Sevil treatment should be examined.

4.     The discussion section is too long, and a big portion of it is just repeat the results.

5.     The materials used in examining the cytokines/chemokines are inappropriate in Figure 7A. Cytokines/chemokines can work in autocrine, paracrine or endocrine manner. The physiologically relevant measurement of them should be cell culture media, not cell lysate. 

6.     In Figure 8, IL10 expression is also highly induced and last for a long period of time. It is not appropriate to state that “the mRNA levels of IL10…was not also observed in RAW264.7 cells after the treatment with SeviL”. 

7.     The manuscript lacks many essential information. For example, the detailed information of Sevil mut, how the monomer is achieved? In figure 9, how long was Sevil treatment? What is the cell lysis buffer composition?

8.     From the data, it is not logical to conclude that PDGFR activation is the cell proliferation mechanism. It could be the accompanying phenomenon. To make this conclusion, at least a knock down experiment is needed.

9.     In line 166-167, G-CSF, in the rest of the manuscript is GM-CSF. They are two different growth factors, and G-CSF is most relevant to neutrophil differentiation. Which one did the authors actually studied? 

Minor:

  1. In Figure 5, the bar for C is missing.
  2. Figure 6 should be reorganized, one column, group MAPKs, then STATs, and also total STATs are suggested.
  3. Line 170, the reference of TNFbeta should be cited.
  4. In Method section, what does 301K, 277K mean?
  5. To compare with macrophage, Hela cell should be classified as “epithelial” not “cervial”.
  6. Within the sentence, it is more common to use Arabic number + units, instead of word + units, eg. 5 mM instead of five mM.
  7. Please give the full description of an abbreviation only the first time it appears, then use the abbreviation afterwards. For example, line 340, iNOS…
  8. Please rewrite lines 290-291, please use “a” (symbol, alpha) instead of “a” in line 171.

Comments on the Quality of English Language

needs minor revision

Author Response

Thank you for reviewing our MS. We gone through your comments and we would like to response for this.

1.In general, M-CSF and GM-CSF are major macrophage differentiation factors; M1, M2 terminology are linked with macrophage polarization status, not their differentiation.

→Thank you for your kind advice. The text was modified, by referring to macrophage polarization as a reprogramming into a new functional phenotype. Several sentenced in the abstract, introduction, and results sections were modified accordingly. The title of the manuscript was also modified to take into account this suggestion.

2. JAK/STAT pathway can be activated by many cytokines, including the most potent anti-inflammatory cytokine IL10. The activation of STATs, especially STAT3, cannot be directly linked to proinflammatory function of Sevil.

→Thank you for this observation. We have removed the mention of the activation of the JAK/STAT pathway from the abstract, better discussing the implications of our findings later in the text.

3. As NFkB is a major player in the expression of proinflammatory cytokines/chemokines, its activity upon Sevil treatment should be examined.

→Thank you for your advice again. We already attempt to find the activation of NFkB of SeviL treated RAW264.7 cells. The activation of NFkB was confirmed by the dissapearance of IkappaBa after the treatment of LPS such as well known positive control. However, this phenomena was not observed in SeviL treated RAW264.7 cells. Identifying the details of cellular mechanism will be the next study.

4. The discussion section is too long, and a big portion of it is just repeat the results.

→Thank you for your kind advice. The discussion section was edited to remove repetitions, unless strictly necessary.

5. The materials used in examining the cytokines/chemokines are inappropriate in Figure 7A. Cytokines/chemokines can work in autocrine, paracrine or endocrine manner. The physiologically relevant measurement of them should be cell culture media, not cell lysate. 

→Thank you for your kind advice. We agree about the fact that using the cell culture media would have been the most appropriate approach to obtain physiologically relevant measurements. However, we have tried the experiments with both types of samples and the results we obtained were not significantly different. Therefore, we chose to keep the results displayed in Figure 7B in their present form.

6. In Figure 8, IL10 expression is also highly induced and last for a long period of time. It is not appropriate to state that “the mRNA levels of IL10…was not also observed in RAW264.7 cells after the treatment with SeviL”. 

→We appreciate for your comments. I changed the sentence present in the results section accordingly.

7. The manuscript lacks many essential information. For example, the detailed information of Sevil mut, how the monomer is achieved? In figure 9, how long was Sevil treatment? What is the cell lysis buffer composition?

→Thank you for this suggestion. I added the required information about SeviL mutants and the details of the treatment whenever appropriate. I also insert the cell lysis buffer composition in part of 4.1 materials.

8. From the data, it is not logical to conclude that PDGFR activation is the cell proliferation mechanism. It could be the accompanying phenomenon. To make this conclusion, at least a knock down experiment is needed.

→I deleted the sentence “These results suggest that the cell proliferation mechanisms induced in macrophage cells by SeviL are related to the activation of PDGFR on the cell surface” which was included in the section 2.3.2, in order to avoid giving the false impression that the activation of PDGFR was directly due to SeviL.

9. In line 166-167, G-CSF, in the rest of the manuscript is GM-CSF. They are two different growth factors, and G-CSF is most relevant to neutrophil differentiation. Which one did the authors actually studied? 

→Thank you for pointing out that. G-CSF is the correct one. We changed all the incorrect mentions of GM-CSF to G-CSF in the text.

Minor:

1.    In Figure 5, the bar for C is missing.

→Thank you for pointing out for this. I inserted scale bar in C.

2. Figure 6 should be reorganized, one column, group MAPKs, then STATs, and also total STATs are suggested.

→Thank you about the suggestion of including an additional total STAT levels in the figure. However, at the present time no antibodies are available in the lab to carry out this task, and the timeframe required by the editorial team for this revision unfortunately would not allow us to include this additional information. Nevertheless, we believe that the information presented concerning the phosphorylated forms of STAT-1 and STAT-3 is the most relevant to investigate their regulation, since only phosphorylated STATs are biologically active.

3.  Line 170, the reference of TNFbeta should be cited.

→Thank you for your suggestion. I added the reference of them.

4. In Method section, what does 301K, 277K mean?

→Thank you for your comments. I rewrite the temperature in Celsius degree instead of Kervin. 310K and 277K indicate approximately 37℃ and 4℃, respectively.

5. To compare with macrophage, Hela cell should be classified as “epithelial” not “cervial”.

→Thank you for your comments. I just indicate the Hela cells as epithelial instead of cervical cells in line 249 (page 9) and line 367 (page 11).

6. Within the sentence, it is more common to use Arabic number + units, instead of word + units, eg. 5 mM instead of five mM.

→Thank you for your suggestion. We adjust the style that you recommended.

7.  Please give the full description of an abbreviation only the first time it appears, then use the abbreviation afterwards. For example, line 340, iNOS…

→Thank you for this suggestion. We carefully checked the text and included full descriptions whenever necessary.

8. Please rewrite lines 290-291, please use “a” (symbol, alpha) instead of “a” in line 171.

→Thank you for your suggestion. I changed the PDFGR-a instead of PDFGR-a in line 173

Reviewer 2 Report

Comments and Suggestions for Authors

The article titled "Multifunctional cell regulation activities of the mussel lectin SeviL: the differentiation into M1 macrophages" authored by Fujii et al. investigates in detail the proliferative effect of the lectin SeviL on rat macrophages (RAW264.7). The authors report interesting results regarding the modulation mechanism (including surface interaction, protein profile response, and phosphorylation) of this lectin on the proliferation of the mentioned cell line. The Introduction is concise and highly informative. The methodology is appropriate for the obtained results, and the discussion addresses all relevant aspects of the work. Therefore, my opinion is favorable to the publication of the paper, with some questions and adjustments described below:

Major:

In the results presented in figure 3 and at other points in the text, the authors compare the effect of mutant SeviL (monomer) and wild-type SeviL (dimer). The methodology does not clearly explain how the mutant was produced or which mutations were induced to produce the monomeric variant. It seems to me that these data are from previous works, and in this study, the dimeric form was used to evaluate the proliferative effect, although in some parts of the manuscript, this is confusing, see lanes 253-256 [The SeviL molecule used in this study was genetically engineered in Escherichia coli, and its activity could be compared with that of monomeric SeviL, in which the amino acids in the polypeptide chain were genetically modified]. Please clarify this aspect in the manuscript.

Minor:

Topic 2.1.1 - Check the text formatting. There are several points where the text is in italics.

Topic 2.3 - Lane 141. 10 ug.mL ?

Topic 2.3.2 - The authors refer to the data of figures 8A and 8B, however, figure 8 is not split into A and B.

Topic 4.5 - Lane 439: [..SeviL has treated RAW264.7 cells several times, respectively...] This sentence is unclear.

Author Response

Thank you for reviewing our MS. We have gone through your helpful comments.

We would like to answer for this.

Major:

In the results presented in figure 3 and at other points in the text, the authors compare the effect of mutant SeviL (monomer) and wild-type SeviL (dimer). The methodology does not clearly explain how the mutant was produced or which mutations were induced to produce the monomeric variant. It seems to me that these data are from previous works, and in this study, the dimeric form was used to evaluate the proliferative effect, although in some parts of the manuscript, this is confusing, see lanes 253-256 [The SeviL molecule used in this study was genetically engineered in Escherichia coli, and its activity could be compared with that of monomeric SeviL, in which the amino acids in the polypeptide chain were genetically modified]. Please clarify this aspect in the manuscript.

→  Thank you for your suggestion. I add some information of monomeric SeviL at the section of material method, 4.3. In addition, I also referred the paper about the monomeric SeviL which we developed in previous work. We used this mutant to compare the function within the wild type.

To avoid the confusion, we revised the sentence by get rid of the part of lane 253-256 in previous.

Minor:

Topic 2.1.1 - Check the text formatting. There are several points where the text is in italics.

 →Thank you for your suggestion. We just corrected the error in line 92-98.

Topic 2.3 - Lane 141. 10 ug.mL ?

 →Thank you for your suggestion. We corrected the error.

Topic 2.3.2 - The authors refer to the data of figures 8A and 8B, however, figure 8 is not split into A and B.

 →Thank you for pointing out the very important issue. Inthe section 2.3.2, we are indicating the protein array data. We now correctly refer to Figure 7B instead of Figure 8B.

Topic 4.5 - Lane 439: [..SeviL has treated RAW264.7 cells several times, respectively...] This sentence is unclear.

→Thank your suggestion. I deleted this sentence.

Round 2

Reviewer 1 Report

Comments and Suggestions for Authors

The revised manuscript (marinedrugs-3002543) by Yuki Fujii et al improved substantially. Here are some remaining concerns:

Major:

1.     My previous comments 3: As NFkB is a major player in the expression of proinflammatory cytokines/chemokines, its activity upon Sevil treatment should be examined.

Authors response: Thank you for your advice again. We already attempt to find the activation of NFkB of SeviL treated RAW264.7 cells. The activation of NFkB was confirmed by the dissapearance of IkappaBa after the treatment of LPS such as well known positive control. However, this phenomena was not observed in SeviL treated RAW264.7 cells. Identifying the details of cellular mechanism will be the next study.

My suggestions: The activity of NFkB is a question can not be ignored when the expression of TNFa and IL6 is substantially changed upon the challenge. The negative data should not be hidden, it should be the origin of new discovery. Even if the authors do not know the underlining mechanisms of Sevil-induced TNFa and IL6 expression, they should mention their findings (negative data can be in supplementary file), and discuss in the Discuss section.

2.     My previous comments 5: The materials used in examining the cytokines/chemokines are inappropriate in Figure 7A. Cytokines/chemokines can work in autocrine, paracrine or endocrine manner. The physiologically relevant measurement of them should be cell culture media, not cell lysate. 

Authors response: Thank you for your kind advice. We agree about the fact that using the cell culture media would have been the most appropriate approach to obtain physiologically relevant measurements. However, we have tried the experiments with both types of samples and the results we obtained were not significantly different. Therefore, we chose to keep the results displayed in Figure 7B in their present form.

My suggestions: It is important to distinguish between statistical difference and quantitative difference. If the authors mean that the data from the media did not reach a statistical difference, then it may only be due to inefficient replicate times. If they mean quantitative difference, they should mention and discuss the result, it could be the wrong measurement time. The shorter incubation time might work, e.g. 3-6 hours instead of 24 hours. The secreted cytokines could be endocytosed by the cells during the 24 hours treatment, making no significant difference between groups.

Minor:

  1. In line 397, please add “nm” after “600”.
  2. Please use "ml" instead of "mL" throughout the manuscript, they are used interchangeably in the manuscript.

Comments on the Quality of English Language

The quality of the English is acceptable. 

Author Response

Major:

1) Reviewer's suggestions: The activity of NFkB is a question can not be ignored when the expression of TNFa and IL6 is substantially changed upon the challenge. The negative data should not be hidden, it should be the origin of new discovery. Even if the authors do not know the underlining mechanisms of Sevil-induced TNFa and IL6 expression, they should mention their findings (negative data can be in supplementary file), and discuss in the Discuss section.

→Thank you for your suggestion. I inserted in a supplementary figure the data that shows the SeviL did not induce the activation NFkappaB, as inferred from the lack of modulation of its key regulator IkappaB. We believe that the most reasonable explanation for this observation is that the activation of cytokine production in this case was mostly due to AP-1, which is positively regulated by the activity of the MAPK cascade. AP-1 and NFkappaB are in fact master regulators of inflammatory responses that, despite being regulated in a different manner, modulate the expression of partly overlapping molecular targets. As a matter of fact, both IL6 (see Hungness et al. 2000, https://doi.org/10.1097/00024382-200014030-00025) and TNF-alpha (see Qiao et al. 2016, https://doi.org/10.1074%2Fjbc.M115.702571) have been demonstrated to be directly regulated also by AP-1. In any case, the interplay between AP-1 and NFkappaB is rather complex (see Fujioka et al. 2004, https://doi.org/10.1128%2FMCB.24.17.7806-7819.2004), since these two transcription factors can also modulate each other, and therefore we cannot exclude that a crosstalk between the two pathways is also involved in this case. These considerations have been added to the discussion.

2) Reviewer's suggestions: It is important to distinguish between statistical difference and quantitative difference. If the authors mean that the data from the media did not reach a statistical difference, then it may only be due to inefficient replicate times. If they mean quantitative difference, they should mention and discuss the result, it could be the wrong measurement time. The shorter incubation time might work, e.g. 3-6 hours instead of 24 hours. The secreted cytokines could be endocytosed by the cells during the 24 hours treatment, making no significant difference between groups.

→ Thank you for your suggestion. I agree that a shorter incubation might be more appropriate for investigating to secretion of cytokines, to study for example IL6 and IL6 receptor interaction. However, this study was focused on the secretion of cytokines mediated by protein-glycan interaction. Compared with the general strength of protein-protein interactions, the binding ability of proteins towards glycan is about approximately 100 times weaker. Considering this factor, we decided to use incubation times longer than 3-6 hours, in order to allow a longer interaction between SeviL and macrophage cells, with the aim to boost its effects. A similar strategy has been previously carried out in other experiments dealing with the evaluation of cytokine production (Yamanishi et al., J Biochem, 2007, https://pubmed.ncbi.nlm.nih.gov/17846063/).

Minor:

  1. In line 397, please add “nm” after “600”.

→Thank you for your kind comment. I added the “nm” after “600”.

  1. Please use "ml" instead of "mL" throughout the manuscript, they are used interchangeably in the manuscript.

→Thank you for your comment. I changed the “ml” instead of “mL” according to your suggestion.

Round 3

Reviewer 1 Report

Comments and Suggestions for Authors

My concerns had been addressed.